# The Molecular Monitoring of an Invasive Freshwater Fish, Brown Trout (*Salmo trutta*), Using Real-Time PCR Assay and Environmental Water Samples

**DOI:** 10.3390/ani15050659

**Published:** 2025-02-24

**Authors:** Su-Hwan Kim, Soo-In Lee, Sang-Hun Lee, So-Eun Jo, Keun-Yong Kim

**Affiliations:** 1Wetland Research Team, National Institute of Ecology, Seocheon 33657, Republic of Korea; ksh0814@nie.re.kr (S.-H.K.); sanghunlee@nie.re.kr (S.-H.L.); 2Invasive Alien Species Team, National Institute of Ecology, Seocheon 33657, Republic of Korea; ecolove093@nie.re.kr; 3Genetic Analysis Team, AquaGenTech Co., Ltd., Busan 48228, Republic of Korea; soeun1013@hanmail.net

**Keywords:** *Salmo trutta*, invasive species, environmental DNA, molecular monitoring

## Abstract

This study presents the development and application of a qPCR assay to monitor an invasive brown trout (*Salmo trutta*) in South Korea using environmental DNA (eDNA). The assay demonstrated high sensitivity and specificity, detecting *S. trutta* in various water samples downstream of the Soyang and Uiam Reservoirs. The results highlight the method’s potential as a non-invasive, efficient, and reliable tool for monitoring invasive fish species and mitigating their ecological impacts.

## 1. Introduction

*Salmo trutta* Linnaeus, 1758, endemic to most of Europe and commonly referred to as brown trout, is a freshwater migratory species that spawns in riverbeds composed of gravel or sand with fast-flowing, cool, and oxygen-rich waters. This species exhibits a high degree of behavioral variability, with some populations being purely freshwater residents, whereas others are anadromous, known as sea trout, and migrate to the ocean before returning to rivers to spawn. This species is prized for its adaptability, thrives in a wide range of habitats, and is well-known for sport fishing [1,2].

Over the past few decades, *S. trutta* has been introduced to many regions outside its native range, including North America, Australia, and parts of Asia such as South Korea, for food consumption and recreational fishing [2,3,4,5,6,7]. As an aggressive predator and competitor, this alien species outcompetes native fish for food and habitat. It preys on native aquatic animals, disrupting natural aquatic ecosystems [2,5,6,7,8,9,10] and also hybridizes with native salmonids, reducing their genetic integrity and fitness, threatening their long-term viability, and destabilizing freshwater ecosystems [11,12,13]. Due to these ecological threats, *S. trutta* is listed among the “100 of the World’s Worst Invasive Alien Species” by the IUCN [14].

Although its introduction has not been recorded in South Korea, Park et al. [8] reported that *S. trutta* successfully established a sexually mature population downstream of the Soyang Reservoir in Chuncheon-si, Gangwon-do, South Korea. This area is enclosed by three reservoirs: Soyang Reservoir, which discharges water into the Soyang River and drains into the northeastern Uiam Reservoir; Paro Lake, which is connected to the Bukhan River and drains into the northwestern Uiam Reservoir; and the southern Uiam Reservoir, which discharges water into the Han River basin. As a result, the *S. trutta* population introduced into South Korea became landlocked. As a cold-water species, its optimal habitat temperature ranges between 12 and 19 °C; however, it is resilient and survives at temperatures up to 24.7 °C [15]. During summer, rivers in South Korea approach 30 °C, limiting the species’ ability to establish a wide distribution. Nevertheless, owing to hydroelectric power generation, the multipurpose dam in the Soyang Reservoir continuously discharges cold water from its mid layer, maintaining a discharge water temperature of 15 °C throughout the year that creates conditions that allow for *S. trutta* survival downstream.

In 2021, the Ministry of Environment of South Korea legally designated *S. trutta* as an invasive species to prevent further spread and minimize ecological damage through eradication and removal projects [16,17]. Owing to its popularity in recreational fishing and ecological adaptability, there are concerns regarding the impact of widespread *S. trutta* release and distribution. Therefore, the close monitoring of its habitat and potential spread is required, necessitating the urgent need for rapid and effective identification in new habitats and areas of expansion. However, the current assessment and monitoring of *S. trutta* are still conducted using traditional methods such as skimming nets, gill nets, or angler-provided rod catch data, as electrofishing—a common method used in freshwater fish surveys—is illegal in South Korea. Given the ecological traits of *S. trutta*, including its preference for cold water and long-distance migration, monitoring through traditional methods is challenging, as these surveys are often expensive, labor intensive, and time consuming.

The environmental DNA (eDNA)-based real-time PCR (quantitative PCR or qPCR) assay is a powerful tool for the molecular monitoring of invasive fish species in aquatic ecosystems, as well as for monitoring commercially exploited, rare, or endangered fish species; for example [18,19,20,21,22,23]. This method involves the collection of water samples from an environment where a target species may be present and an analysis of the obtained samples for the detection of biological material, such as skin cells, scales, mucus, blood, feces, and gametes, shed by the species. qPCR amplifies DNA fragments unique to the target species, enabling highly sensitive detection even at low population densities. This technique is particularly useful for early detection, tracking the spread, and monitoring the distribution of exotic fish species, such as *S. trutta* [22,24,25,26,27].

In this study, we aimed to determine the distribution range of *S. trutta* through a qPCR analysis of water samples collected from river water downstream of the Soyang Reservoir and around the Uiam Reservoir, where the species is known to occur.

## 2. Materials and Methods

### 2.1. Sampling of Fish Specimens and Genomic DNA (gDNA) Extraction

Specimens of *S. trutta* (*n* = 2), the other five salmonid species, i.e., *Brachymystax lenok tsinlingensis*, *Oncorhynchus keta*, *Oncorhynchus mykiss*, *Thymallus grubii*, and *Salmo salar*, and 19 freshwater fish species belonging to diverse orders and families (*Acheilognathus rhombeus*, *Coreoperca herzi*, *Cyprinus carpio*, *Gasterosteus aculeatus*, *Hemiculter eigenmanni*, *Lepomis macrochirus*, *Liobagrus andersoni*, *Misgurnus anguillicaudatus*, *Monopterus albus*, *Mugil cephalus*, *Odontobutis platycephala*, *Orthrias nudus*, *Oryzias latipes*, *Plecoglossus altivelis*, *Repomucenus olidus*, *Rhynchocypris oxycephalus*, *Silurus asotus*, *Tachysurus fulvidraco*, and *Zacco platypus*) were collected from rivers or local markets in South Korea. A piece of the pelvic fin of each species was excised and used for gDNA extraction, as previously described by Asahida et al. [28]. The extracted gDNA was resuspended in TE buffer (10 mM Tris-HCl and 1 mM EDTA, pH 8.0), and its quantity and quality were assessed using a spectrophotometer (NanoDrop™ One, Thermo Fisher Scientific Inc., Waltham, MA, USA).

### 2.2. Modification of Primers and Probe

All mitochondrial cytochrome *b* gene (*mt-cyb*) sequences of *S. trutta* haplotypes and all reference sequences of the mitochondrial genomes of salmoniform species were retrieved from GenBank (https://www.ncbi.nlm.nih.gov/; accessed on 17 July 2023) and aligned using CLUSTALW, implemented in BioEdit 7.2 [29]. Following a comparison of the aligned nucleotide matrices, the forward and reverse primers and hydrolysis probe reported by Carim et al. [26] were modified and used in the qPCR assays. Their melting temperature (*T*_m_) and secondary structures were predicted using the Sequence Manipulation Suite ver. 2 (https://www.bioinformatics.org/sms2/; accessed on 17 July 2023) [30] and optimized prior to oligonucleotide syntheses.

### 2.3. Environmental Water Sampling and eDNA Extraction

Environmental water samples (each 1–2 L) were collected in sterile disposable plastic bottles at a depth of 10 m from eight stations located downstream of the Soyang Reservoir and around the Uiam Reservoir between January and March 2023 (Table 1) in Chuncheon-si, Gangwon-do, South Korea. The samples were vacuum-filtered through glass microfiber filters (Grade GF/F circles, 47 mm; Whatman, Marlborough, MA, USA). Each filter was folded in half four times using uncontaminated forceps, inserted into a 2.0 mL microtube, and placed in a cooled ice box with polyethylene bags containing ice, protected from light exposure. They were then transported directly to the laboratory and stored in at −70 °C until further use. Each filter was broken using Omni Bead Ruptor 12 Bead Mill Homogenizer (OMNI International, Kennesaw, GA, USA) and eDNA was extracted using DNeasy Blood & Tissue kit (Qiagen, Hilden, Germany), following the manufacturer’s instructions. The eDNA was finally eluted in 50 μL of sterile distilled water and immediately stored at –20 °C for further processing.

### 2.4. qPCR Assay

qPCR amplification was conducted in triplicate (three technical replicates) for each eDNA sample using GoTaq^®^ Probe qPCR Master Mix (Promega, Madison, WI, USA). The reaction mixture (10 μL) contained 5 μL of GoTaq^®^ Probe qPCR Master Mix, 1 μL of CXR Reference Dye, 1 μL of eDNA as a template, and 0.2 μM each of the forward primer Str-cyb-0297f (5′-CCGAGGACTCTACTATGGT-3′) and the reverse primer Str-cyb-0384r (5′-GGAAGAACGTAGCCCACG-3′) and the hydrolysis probe Str-cyb-0341p (5′-FAM-ATATCGGAGTCGTACTGCTA-MGB-Eclipse-3′), which were synthesized by Macrogen, Inc. (Seoul, South Korea). qPCR was performed using a QuantStudio™ 5 Real-Time PCR System (Thermo Fisher Scientific Inc.) with the following cycling conditions: an initial activation at 95 °C for 2 min, followed by 50 cycles of denaturation at 95 °C for 15 s, and annealing and extension at 60 °C for 45 s. The gDNAs (20 ng μL^−1^) extracted from two *S. trutta* specimens were used as positive controls. Sterile distilled water was used as a negative control to monitor contamination during filtration, eDNA extraction, and qPCR analysis.

A synthesized partial DNA fragment of *S. trutta mt-cyb* containing all the binding sites for the forward and reverse primers and the hydrolysis probe was produced by gene synthesis and inserted into a plasmid by Bioneer Inc. (Daejeon, South Korea). The sensitivity of the primers and probe was tested by qPCR assay against 10-fold serial dilutions of the plasmid DNA (10^9^ copies rxn^−1^) in triplicate for each dilution. The results were used to produce a standard curve to quantify the copy numbers of *S. trutta* eDNA. Their specificity was also tested against five other salmonid and nineteen freshwater fish species belonging to diverse orders and families that are commonly found in South Korea.

## 3. Results

### 3.1. Modified Primers and Probe

In this study, we modified the forward and reverse primers, as well as the hydrolysis probe, previously described by Carim et al. [26], which were specific to *S. trutta* and designed based on *mt-cyb* sequences (Table 2). The *T*_m_ of the forward and reverse primers used in this study (Str-cyb-0297f and Str-cyb-0384r, respectively) were lower than those used by Carim et al. [26] (Str-cyb-0294f and Str-cyb-0382r, respectively). Both primers used in the present study showed the exact matches with all *S. trutta* haplotypes in the GenBank database. Additionally, the hydrolysis probe used in this study (Str-cyb-0341p) had a higher *T*_m_ than that used by Carim et al. [26] (Str-cyb-0345p). The probe also matched all *S. trutta* haplotypes in the GenBank database, with a single base-pair mismatch in only one haplotype (GenBank accession number JX960839) reported by Crête-Lafrenière et al. [31], but multiple base-pair mismatches with other salmoniform species, including the congeneric *Salmo ischchan*, *Salmo obtusirostris*, and *S. salar*.

The forward and reverse primers produced a 105-bp amplicon, as predicted using conventional PCR amplification. The hydrolysis probe consisted of a 20-mer oligonucleotide with a fluorophore, 6-carboxyfluorescein (6-FAM), covalently attached to the 5′-end, and a quencher, MGB-Eclipse, at the 3′-end. This combination of oligonucleotides was unique to *S. trutta* and was not found in other salmoniform species.

### 3.2. Sensitivity and Specificity Tests

qPCR amplification using the designed primers and probe resulted in positive amplification with a quantification cycle (C_q_) value of 17.776 against a standard concentration of the gDNA (20 ng μL^−1^) extracted from *S. trutta*. No background amplification was observed. The sensitivity test was conducted using a serial dilution of plasmid DNA (1–10^9^ copies rxn^−1^), in which a partial DNA fragment of *S. trutta mt-cyb* containing all the binding sites for the forward and reverse primers and the hydrolysis probe were inserted. An inverse relationship was observed between the C_q_ values and plasmid DNA concentrations, with a detection limit of as low as 10^2^ copies rxn^−1^ of plasmid DNA and a C_q_ value of 36.717 (Figure 1). Therefore, the C_q_ value was set as the lowest detection limit to achieve acceptable levels of precision and accuracy in the qPCR assay. This correlation was used to generate the standard curve slope, which produced the following linear regression equation:y = −3.51x + 43.818 (*r*^2^ = 0.999, efficiency = 92.69%)

Linear regression was used to detect and quantify *S. trutta* eDNA in environmental water samples.

To confirm the specificity of the primers and hydrolysis probe for *S. trutta*, we carried out qPCR amplification of five salmonid (*B. lenok tsinlingensis*, *O. keta*, *O. mykiss*, *T. grubii*, and *S. salar*) and nineteen freshwater fish species belonging to diverse orders and families that are commonly found in the rivers and lakes or local markets of South Korea. Only two *S. trutta* specimens successfully produced positive fluorescence amplifications, whereas the other fish species failed to produce measurable amplifications (Figure 2).

### 3.3. qPCR Assay of Environmental Water Samples

The primers and hydrolysis probe specific to *S. trutta* in this study were used to amplify eDNA extracted from the collected environmental water samples (*n* = 24). Positive amplifications were observed in 11 samples (Stns. 1, 2, and 8 in January 2023; Stns. 1, 2, 5, and 8 in February 2023; and Stns. 1, 6, 7, and 8 in March 2023), with C_q_ values ranging from 30.665 to 38.629, corresponding to plasmid copy numbers from 250.6 copies L^−1^ to 132,677.1 copies L^−1^ (Table 3). The highest value, 132,677.1 ± 6386.3 copies L^−1^, was observed for St. 1.

*S. trutta* eDNA was detected in all three replicates in the upstream section of the Soyang River (St. 1), located downstream of the Soyang Reservoir, between January and March 2023 (Figure 3). It was also detected in all three replicates from the downstream section (St. 8) in February 2023 but in only one replicate out of three in January and March. Additionally, it was detected in all three replicates from Sangjung Island (St. 2), located upstream of the Uiam Reservoir, in January and February 2023; however, no *S. trutta* eDNA was detected in March. In contrast, two small streams flowing into the Uiam Reservoir (Stns. 5 and 6) possessed *S. trutta* eDNA in only one out of three replicates in February and March, whereas St. 7, located downstream of the Uiam Reservoir, showed *S. trutta* eDNA in only one of the three replicates in March at very low concentrations. However, *S. trutta* eDNA was not detected in the Gongji Stream (St. 3), which flows into the Uiam Reservoir, or in St. 4, which is located downstream of the Bukhan River. Representative samples that showed the positive amplifications were further verified via Sanger sequencing to confirm the absence of false-positive results.

## 4. Discussion

An eDNA-based qPCR assay enhances management efforts aimed at preventing the establishment and spread of invasive species, and by providing rapid and precise results, is useful in protecting native biodiversity [32,33]. The effectiveness of qPCR for species detection depends on the development of species-specific forward and reverse primers and/or hydrolysis probe that exclusively amplify the DNA fragment of the target species, minimizing false positives from cross-amplification with closely related species [34,35]. To ensure target specificity, the primers and/or probe need to be tested against all related species potentially present in the study area to confirm that only the target species is amplified. This validation process is essential for its accurate detection using molecular monitoring. Using the primers and probe specific to *S. trutta* that were modified from Carim et al. [26], we aimed to amplify *S. trutta* eDNA from environmental water samples by qPCR assay, even if present at low concentrations, to allow for its accurate and early detection.

It has been previously reported that qPCR targeting *S. trutta* is an important tool for detecting and monitoring this invasive species [22,25,26,27]. This method aids in tracking the spread of *S. trutta* and supports efforts to control its population and mitigate ecological damage. In this study, we modified the forward and reverse primer of Carim et al. [26] by reducing their *T*_m_ and the hydrolysis probe by increasing its *T*_m_ to enhance qPCR sensitivity and specificity. Reducing the *T*_m_ of both primers improved their binding efficiency to eDNA at low temperatures, increasing the amplification efficiency, especially in low-concentration samples, and resulting in greater detection sensitivity. Additionally, increasing the *T*_m_ of the hydrolysis probe enhanced its binding stability to the target sequence, ensuring more selective binding to the correct target and reducing nonspecific binding. This minimizes false positives and enables a more accurate detection of the target species. Aligning the *T*_m_ of both primers and probe improved the overall balance and efficiency of the qPCR, thus enhancing the robustness of the qPCR assay, particularly when complex or degraded samples are used. These optimizations led to a more reliable and precise detection of the target species, such as *S. trutta*, via molecular monitoring.

In the present study, *S. trutta* eDNA was detected in high quantities and frequencies in the upstream (St. 1) and downstream (St. 8) sections of the Soyang River, which is consistent with previous findings [7,17,36,37]. These areas are characterized by shallow water depths, with riverbeds mainly composed of sand, pebbles, gravel, and boulders [8,17]. In addition to these two stations, *S. trutta* eDNA was detected on Sangjung Island (St. 2), located upstream of the Uiam Reservoir. The environmental water was released from the middle layer of the Soyang Reservoir for hydroelectric power generation, maintaining a discharge water temperature of 15 °C throughout the year. As a result, the downstream area, approximately 10 km before merging with the Bukhan River, is influenced by the cold water released from the reservoir [38]. These three stations were directly affected by such cold water. Thus, the riverbed structure and hydraulic characteristics of the Soyang River provide an optimal environment for *S. trutta* inhabitation and spawning [39,40,41], because this species requires low water temperatures and high levels of dissolved oxygen. Park et al. [8] observed sexually mature males and females in the Soyang Reservoir, supporting the conclusion that *S. trutta* migrates downstream from the Soyang Reservoir for spawning. Therefore, these areas serve as overwintering habitats for anadromous fish, such as *S. trutta*, which migrate to the upper rivers or streams for reproduction. This species also has a high potential to expand its habitat to the Paro Reservoir, located upstream of the Uiam Reservoir, because of its distinct life cycle, necessitating broader monitoring efforts across the Han River basin.

Our study confirmed previously known habitats of *S. trutta* and demonstrated that this fish is expanding its habitat. Consistent with previous findings, we found that *S. trutta* eDNA was detected in the Soyang River [8,37]; however, we also detected *S. trutta* in the Uiam Reservoir, indicating that the species utilizes a broader range of habitats than previously reported. Although the quantity of eDNAs does not always directly reflect the biomass or number of individuals, owing to various biotic and abiotic factors in the natural habitats of the target species [21,27,42,43,44], our results highlight qPCR as an alternative tool for determining the presence of *S. trutta* in aquatic environments. This method offers a more efficient and effective approach for assessing and monitoring practices than traditional field surveys. Furthermore, this approach can support conservation efforts by habitats that require the removal of invasive species with the goal of restoring the area to its predefined historical conditions [25].

This survey was conducted during a limited period, specifically during the winter months (January and February) and early spring (March), when water was at the lowest annual temperature [8]. This water condition is suitable for *S. trutta*, enabling broad species distribution. It is necessary to apply molecular monitoring during warmer months, that is, from late spring to fall, when water temperatures rise. During this period, it is likely that *S. trutta* moves to deeper and cooler areas at the bottom of the Uiam Reservoir, where a lower water temperature is maintained. In this study, we found that *S. trutta* eDNA was more frequently detected at the southern stations (Stns. 5, 6, and 7), which were located around the main water body of the Uiam Reservoir in February and March, when the water temperature was higher than that in January.

In addition, *S. trutta* frequently hybridizes with other native salmonid species [12,13]. It also has a high potential to hybridize with native salmonid species in South Korea, such as *B. lenok tsinlingensis* and *O. masou masou*, leading to genetic mixing that reduces the integrity and fitness of native populations. Such hybridization may result in the loss of locally adapted traits, inhibiting resilience to environmental changes, climate stress, and competition with invasive salmonid species. The introduction of *S. trutta* through intentional stocking or natural migration to other rivers and streams in South Korea could negatively impact the natural genetic pollution, further endangering native populations. *S. trutta* fishing has gained popularity, and it was reported in online communities that attempts have been made to transplant this species into other rivers or streams outside the Uiam Reservoir, highlighting the need for ongoing monitoring to prevent its spread. Overall, our findings highlight the potential for the future use of our qPCR assay for the monitoring of invasive fish species.

## 5. Conclusions

This study demonstrated that the qPCR assay using the species-specific primers and probe is effective for the molecular monitoring of an invasive *S. trutta* in South Korea. The assay successfully detected *S. trutta* eDNA from environmental water samples collected downstream of the Soyang Reservoir and around the Uiam Reservoir, even at low concentrations. Our findings confirm the presence of *S. trutta* in these areas and highlight its expanding habitat range, raising concerns about potential ecological impacts on native freshwater species. The qPCR assay developed in this study provides a reliable, sensitive, and efficient tool for the early detection and continuous monitoring of *S. trutta*, offering an effective alternative to traditional sampling methods. Continuous molecular monitoring is essential to prevent further spread, mitigate ecological risks, and support conservation efforts aimed at protecting native biodiversity in South Korean freshwater ecosystems.

## Figures and Tables

**Figure 1 animals-15-00659-f001:**
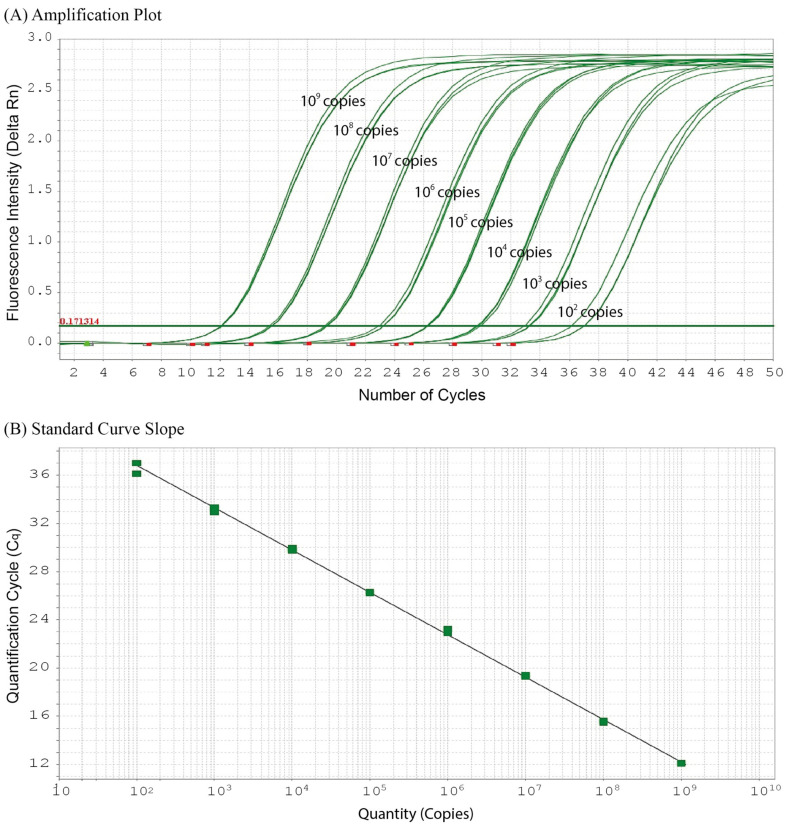
Amplification plot and standard curve of the real-time PCR assay using the primers and hydrolysis probe specific to brown trout, *Salmo trutta*. A serial dilution of plasmid DNA (1–10^9^ copies rxn^−1^) shows a detection limit of as low as 10^2^ copies rxn^−1^ (**A**), with the standard-curve slope producing a linear regression (**B**).

**Figure 2 animals-15-00659-f002:**
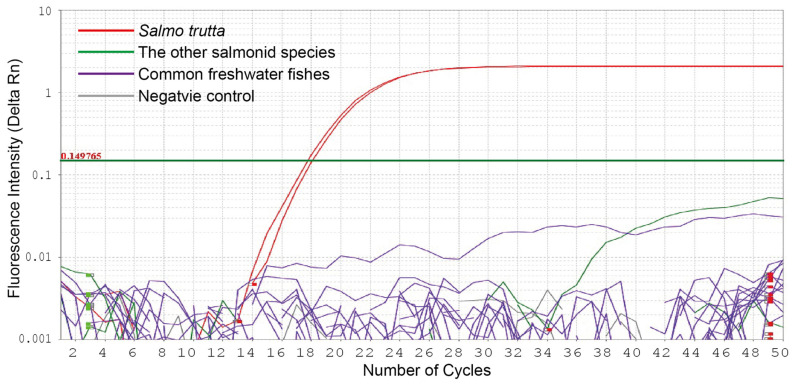
An amplification curve of real-time PCR assay using the primers and hydrolysis probe specific to brown trout, *Salmo trutta.* Specificity testing was conducted on five salmonid species, *Brachymystax lenok tsinlingensis*, *Oncorhynchus keta*, *Oncorhynchus keta*, *Thymallus grubii*, and *Salmo salar*, as well as two *S. trutta* specimens and nineteen freshwater fish species commonly found in South Korean rivers and lakes or local markets.

**Figure 3 animals-15-00659-f003:**
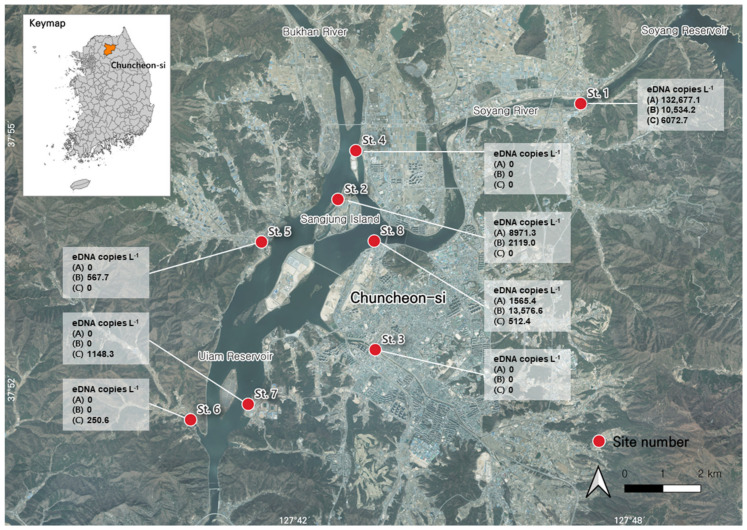
Average environmental DNA (eDNA) concentrations (copies L^−1^) of brown trout, *Salmo trutta*, quantified using the real-time PCR assay across eight stations in the downstream river of the Soyang Reservoir and around Uiam Reservoir. (A) January 2023; (B) February 2023; and (C) March 2023.

**Table 1 animals-15-00659-t001:** Sampling sites of environmental water samples for the molecular monitoring of brown trout, *Salmo trutta*, downstream of the Soyang Reservoir and around the Uiam Reservoir in Chuncheon-si, Gangwon-do, South Korea.

Station	GPS Coordinate	Location
St. 1	37°55′37.17″ N 127°47′07.03″ E	Upstream of the Soyang River
St. 2	37°54′15.93″ N 127°42′46.12″ E	The Sangjung Island
St. 3	37°52′08.34″ N 127°43′26.22″ E	The Gongji Stream
St. 4	37°54′57.30″ N 127°43′05.38″ E	Downstream of the Bukhan River
St. 5	37°53′39.78″ N 127°41′23.97″ E	Small stream flowing into the Uiam Reservoir
St. 6	37°51′08.81″ N 127°40′07.78″ E	Small stream flowing into the Uiam Reservoir
St. 7	37°51′21.98″ N 127°41′09.69″ E	Downstream of the Uiam Reservoir
St. 8	37°53′40.60″ N 127°43′25.11″ E	Downstream of the Soyang River

**Table 2 animals-15-00659-t002:** A comparison of the forward and reverse primers and the hydrolysis probe from Carim et al. [29] and those modified in this study, which are specific to brown trout, *Salmo trutta*.

Oligonucleotide Name ^1^	Sequence (5′ → 3′)	G + C (%)	Nearest Neighbor *T*_m_ (°C)	References
Forward primer				
Str-cyb-0294f	CGCCCGAGGACTCTACTATGGT	59.09	67.87	Carim et al. [26]
Str-cyb-0297f	CCGAGGACTCTACTATGGT	52.63	60.07	This study
Reverse primer				
Str-cyb-0382r	GGAAGAACGTAGCCCACGAA	55.00	65.00	Carim et al. [26]
Str-cyb-0384r	GGAAGAACGTAGCCCACG	61.11	62.94	This study
Hydrolysis probe				
Str-cyb-0345p	CGGAGTCGTACTGCTAC	58.82	58.83	Carim et al. [26]
Str-cyb-0341p	ATATCGGAGTCGTACTGCTA	45.00	60.01	This study

^1^ The oligonucleotides were named based on the species name, with the first letter of the genus (S) and the first two letters of the species (tr) for *S. trutta*, followed by the gene name, mitochondrial cytochrome *b* gene (*mt-cyb*), the relative position of the first base of each oligonucleotide from the start codon of *mt-cyb*, and their direction: forward (f); reverse (r); or function probe (p).

**Table 3 animals-15-00659-t003:** The quantification of environmental DNA from water samples collected downstream of the Soyang Reservoir and around Uiam Reservoir using real-time PCR analysis with the primers and hydrolysis probe specific to brown trout, *Salmo trutta*.

Water Sample	Volume (mL)	Quantification Cycle (C_q_) Value			Total Volume Equivalent (Copies L^−1^)			Average	Standard Deviation
Replicate		1	2	3	1	2	3		
January 2023									
St. 1	2000	30.757	30.809	30.665	131,416	127,015	139,600	132,677.1	6386.3
St. 2	2000	34.607	34.628	35.457	10,517	10,375	6022	8971.3	2555.0
St. 3	2000	ND ^1^	ND	ND	0	0	0	0.0	0.0
St. 4	2000	ND	ND	ND	0	0	0	0.0	0.0
St. 5	2000	ND	ND	ND	0	0	0	0.0	0.0
St. 6	2000	ND	ND	ND	0	0	0	0.0	0.0
St. 7	2000	ND	ND	ND	0	0	0	0.0	0.0
St. 8	2000	35.836	ND	ND	4696	0	0	1565.4	2711.3
February 2023									
St. 1	2000	34.116	34.961	34.886	14,510	8335	8758	10,534.2	3449.3
St. 2	2000	36.464	36.399	ND	3111	3246	0	2119.0	1836.4
St. 3	2000	ND	ND	ND	0	0	0	0.0	0.0
St. 4	2000	ND	ND	ND	0	0	0	0.0	0.0
St. 5	2000	37.382	ND	ND	1703	0	0	567.7	983.2
St. 6	2000	ND	ND	ND	0	0	0	0.0	0.0
St. 7	1000	ND	ND	ND	0	0	0	0.0	0.0
St. 8	2000	35.039	34.237	33.673	7920	13,408	19,402	13,576.6	5742.7
March 2023									
St. 1	2000	35.469	34.946	36.150	5976	8420	3822	6072.7	2300.9
St. 2	2000	ND	ND	ND	0	0	0	0.0	0.0
St. 3	2000	ND	ND	ND	0	0	0	0.0	0.0
St. 4	2000	ND	ND	ND	0	0	0	0.0	0.0
St. 5	2000	ND	ND	ND	0	0	0	0.0	0.0
St. 6	2000	ND	ND	38.629	0	0	752	250.6	334.1
St. 7	1000	37.365	ND	ND	3445	0	0	1148.3	1989.0
St. 8	2000	ND	ND	37.539	0	0	1537	512.4	887.5
Negative control	1000	ND	ND	ND	0	0	0	0.0	0.0

^1^ ND: not detected.

## Data Availability

The original contributions presented in this study are included in the article.

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
