# Peer review of "The Molecular Monitoring of an Invasive Freshwater Fish, Brown Trout (Salmo trutta), Using Real-Time PCR Assay and Environmental Water Samples"

_animals, 2025, doi:10.3390/ani15050659_

Round 1

Reviewer 1 Report

Comments and Suggestions for Authors

Review for manuscript with ID (animals-3487124)
Q1. The authors should follow the journal guidelines for writing references and citations in text.
Q2. Table 2 should be supported by several data such as NCBI GenBank accession numbers, primer efficiency, product size (pb), R2 and Pearson’s coefficient.

Author Response

Comment 1: The authors should follow the journal guidelines for writing references and citations in text.

Respond 1: Thank you for pointing this out. We reformatted references in text according to the journal guidelines of 'Aquatic Animals.' Please look the corrections throughout the revised manuscript.

Comment 2: Table 2 should be supported by several data such as NCBI GenBank accession numbers, primer efficiency, product size (pb), R2 and Pearson’s coefficient.

Respond 2: Table 2 provides only oligonucleotide sequences of primers and probe. Because of their short lengths it is not allowed to register in GenBank. 'Sensitivity and specificity tests' subsection of the 'Result' section described primer efficiency, product size (bp), r2 value and Pearson’s coefficient in detail used in this study. Please see the lines 179-221.

Reviewer 2 Report

Comments and Suggestions for Authors

The present work, utilizing environmental DNA (eDNA) from river water samples collected downstream of the Soyang Reservoir and around the Uiam Reservoir, aims to apply a sensitive and specific qPCR assay using primers and a hydrolysis probe targeting the mitochondrial cytochrome b (mt-cyb) gene of S. trutta. This study demonstrates the effectiveness of the qPCR assay for detecting S. trutta in aquatic environments and highlights its potential for monitoring the spread of this species, particularly in areas that are difficult to survey using traditional methods. This is a valuable piece of scientific work, conducted rigorously using a modern molecular approach, providing an efficient tool for S. trutta population management and helping to mitigate its impact on native biodiversity.

I just have minor suggestions and comments for the authors:

Abstract: The abstract can be enhanced to more effectively emphasize the value and significance of the work.

Lines 13-14: “The Ministry of Environment of South Korea has legally designated S. trutta as an invasive species.” I suggest removing this sentence.

Line 26: “Keywords: environmental DNA; invasive species; molecular monitoring; Salmo trutta” I recommend “Keywords: Salmo trutta; invasive species; environmental DNA; molecular monitoring.

Introduction:

The Introduction can be summarized or condensed.

Lines 29-38: “Salmon and trout are characterized by ............ in Korean riverine ecosystems and are both 37 culturally and commercially important.”: This paragraph can be removed to focus exclusively on presenting Salmo trutta.

Line 55: The reference to 'Verspoor 1988' can be removed, as it is outdated.

Lines 45-61:“Over the past few decades, S. trutta has been ............ of the World’s Worst Invasive Alien Species” (Lowe et al. 2000).” This paragraph is highly interesting but could be summarized or condensed.

Lines 82-84: ” However, the current assessment and  monitoring ............is illegal in South Korea.” Please merge the sentences for better flow and coherence.

Lines 85-87: Given ecological traits of S. trutta, including its preference for cold water and long-distance migration, monitoring via traditional methods is challenging. Moreover, these surveys are expensive, labor-intensive, and time-consuming.” Please merge the sentences to improve flow and coherence and summarize the content concisely.

Materials and methods:

Line 104: “Materials and Methods” I recommend “Materials and methods” .

Line 107 : Which fish species “19 common freshwater fish species”?!

Line 116: “using ClustalW in BioEdit” I recommended “using CLUSTALW, implemented in BioEdit 7.2”

Line 137: Table 1.: the table form can be improved.

The location names could be revised to be more concise.

Results:

Please try to place the tables or figures close to the main text where they are referenced.

Line 167: “Salmo ischchan, Salmo obtusirostris, and Salmo salar.”: I recommend “S. ischchan, S. obtusirostris, and S. salar” .

Line 182: Table 2: “Reference” I recommend “References” / Please delete “Sequence length (bp)” column/ Delete primers underlines/ Replace “GC content (%)” by “ G+C(%)”

Discussion:

The Discussion is quite interesting but could be improved further.

Lines 275- 279: “As the habitat range of S. trutta, ............ are necessary for effective management practices. “This has already been cited in the introduction. I recommend removing it to avoid redundancy.

Line 333: different font styles along the sentence.

I suggest introducing “a Conclusion section”.

Comments on the Quality of English Language

The English could be improved to more clearly express the research.

Author Response

We sincerely thank the reviewer for the thoughtful and constructive comments that helped improve the quality and clarity of our manuscript. Your valuable feedback guided us in refining our work, and we appreciate the time and effort you dedicated to reviewing our paper.

Comment 1: Lines 13-14: “The Ministry of Environment of South Korea has legally designated S. trutta as an invasive species.” I suggest removing this sentence.

Respond 1: According to the suggestion the sentence was deleted in the Abstract section.

Comment 2: Line 26: “Keywords: environmental DNA; invasive species; molecular monitoring; Salmo trutta” I recommend “Keywords: Salmo trutta; invasive species; environmental DNA; molecular monitoring”.

Respond 2: The Keywords were changed according to the suggestion. Please see lines 24-25.

Comment 3: “Salmon and trout are characterized by ............ in Korean riverine ecosystems and are both 37 culturally and commercially important.”: This paragraph can be removed to focus exclusively on presenting Salmo trutta.

Respond 3: According the suggestion we deleted the paragraph.

Comment 4: Line 55: The reference to 'Verspoor 1988' can be removed, as it is outdated.

Respond 4: According the suggestion we deleted the reference.

Comment 5: Lines 45-61: “Over the past few decades, S. trutta has been ............ of the World’s Worst Invasive Alien Species” (Lowe et al. 2000).” This paragraph is highly interesting but could be summarized or condensed.

Respond 5: According the suggestion we condensed the paragraph. Please see the lines 56-63.

Comment 6: Lines 82-84: ” However, the current assessment and monitoring ............is illegal in South Korea.” Please merge the sentences for better flow and coherence.

Respond 6: According the suggestion we merged the sentences. Please see the lines 83-86.

Comment 7: Lines 85-87: Given ecological traits of S. trutta, including its preference for cold water and long-distance migration, monitoring via traditional methods is challenging. Moreover, these surveys are expensive, labor-intensive, and time-consuming.” Please merge the sentences to improve flow and coherence and summarize the content concisely.

Respond 7: According the suggestion we merged the sentences. Please see the lines 89-92.

Comment 8: Line 104: “Materials and Methods” I recommend “Materials and methods”.

Respond 8: According the suggestion we corrected it. Please see line 108.

Comment 9: Line 107: Which fish species “19 common freshwater fish species”?!

Respond 9: We specified the “19 common freshwater fish species.” Please see the lines 111-116.

Comment 10: Line 116: “using ClustalW in BioEdit” I recommended “using CLUSTALW, implemented in BioEdit 7.2”

Respond 10: We corrected it according to the recommendation.

Comment 11: Line 137: Table 1.: the table form can be improved. The location names could be revised to be more concise.

Respond 11: According the suggestion we revised the table in more concise.

Comment 12: Results: Please try to place the tables or figures close to the main text where they are referenced.

Respond 12: According the suggestion we relocated the Figure 3.

Comment 13: Line 167: “Salmo ischchan, Salmo obtusirostris, and Salmo salar.”: I recommend “S. ischchan, S. obtusirostris, and S. salar”.

Respond 13: Because the the species were mentioned for the first time in the manuscript we used their full genus name. But Salmo salar we corrected it as S. salar because of its previous use of the full species name.

Comment 14: Line 182: Table 2: “Reference” I recommend “References” / Please delete “Sequence length (bp)” column/ Delete primers underlines/ Replace “GC content (%)” by “ G+C(%)”

Respond 14: According the suggestion we corrected Table 2. Please see lines 187-192.

Comment 15: Lines 275- 279: “As the habitat range of S. trutta, ............ are necessary for effective management practices. “This has already been cited in the introduction. I recommend removing it to avoid redundancy.

Respond 15: According the recommendation we deleted this paragraph.

Comment 16: I suggest introducing “a Conclusion section”.

Respond 16:  According the suggestion we introduced the "Conclusion" section. Please see the lines 380-391.

Reviewer 3 Report

Comments and Suggestions for Authors

Dear Editor,

This study highlights the effectiveness of eDNA-based qPCR assays in monitoring Salmo trutta populations in South Korea. The modified primers and hydrolysis probe exhibited high specificity and sensitivity, successfully amplifying S. trutta DNA from environmental water samples. The qPCR results provide crucial insights into the species’ spatial and temporal distribution within the monitored locations.

Comments and Suggestions:

  • Line 33: The reference "Groot and Margolis 1991" is in bold. Please ensure consistent formatting with other references.

  • Line 33: The authors mention 11 species, including three endemic and one introduced. What about the remaining species? Providing details on all mentioned species would enhance clarity.

  • Line 39: When referring to Salmo trutta Linnaeus, 1758 for the first time, a citation indicating the original taxonomic description should be included.

  • Line 45: Ensure consistency in the formatting of in-text citations (e.g., "Klemetsen et al. 2003" vs. "Kim et al. 2005").

  • Line 50: Some sentences are long and complex, making comprehension difficult. Breaking them into shorter, more direct sentences would improve readability.

    • Example: "As an aggressive predator and competitor, this alien species outcompetes native fish for food and habitat. It also preys on native aquatic animals, disrupting natural aquatic ecosystems."

  • Line 62: The section states that "exact records of its introduction are unavailable." Providing more details on potential introduction pathways (e.g., aquaculture escape, illegal stocking) would strengthen the discussion.

  • Line 69: The temperature discussion appears somewhat contradictory. It states that most South Korean rivers approach 30°C in summer, which should hinder S. trutta establishment. However, the role of hydroelectric dam discharge in sustaining the species should be expanded upon to clarify how S. trutta persists in such conditions.

  • Line 113: While the authors modified existing primers, further validation steps, such as in silico specificity tests and empirical sensitivity assessments, should be included.

  • Line 143: When describing the PCR mix, use "concentrations" instead of generic descriptions. Additionally, specify the amount of Taq polymerase (U) and buffer in the reaction mix.

  • Line 221: The study provides detailed modifications of primers and probes, but a deeper discussion on the rationale behind these changes and how they improve efficiency over previous designs would enhance clarity.

    • Did the authors conduct an in silico analysis to assess specificity before empirical testing?

  • Line 248: The results indicate fluctuations in S. trutta eDNA across different sites and months. Expanding on potential environmental factors influencing these variations (e.g., water temperature, flow rate, degradation rates) would strengthen ecological insights.

    • Some sites showed no S. trutta eDNA despite the species' known presence in the region. Addressing potential reasons for false negatives, such as eDNA degradation, dilution, or inhibition, would provide a more nuanced understanding of assay limitations.

  • Line 280: Further explanation of how the detection limit relates to species abundance and ecological monitoring would be beneficial.

    • Analyzing potential factors such as water temperature, flow rate, and eDNA degradation on detection patterns would strengthen ecological interpretations.

    • Some locations lacked S. trutta eDNA despite known presence. Discussing potential causes such as sample degradation, dilution, or PCR inhibition would improve understanding of assay limitations.

    • Incorporating this qPCR-based approach into continuous monitoring programs would enhance its practical application.

Overall, this study presents a valuable contribution to invasive species monitoring using eDNA. Addressing these points would further strengthen the manuscript and its applicability to conservation and management efforts.

Author Response

We sincerely thank the reviewer for the thoughtful and constructive comments that helped improve the quality and clarity of our manuscript. Your valuable feedback guided us in refining our work, and we appreciate the time and effort you dedicated to reviewing our paper.

Comment 1: Line 33: The reference "Groot and Margolis 1991" is in bold. Please ensure consistent formatting with other references.

Respond 1: The references in the manuscript were reformatted according to the journal guideline throughout this revised manuscript.

Comment 2: Line 33: The authors mention 11 species, including three endemic and one introduced. What about the remaining species? Providing details on all mentioned species would enhance clarity.

Respond 2: The paragraph that included the sentences were deleted according to the other reviewer's suggestion.

Comment 3: Line 39: When referring to Salmo trutta Linnaeus, 1758 for the first time, a citation indicating the original taxonomic description should be included.

Respond 3: According to the comment we included the citation. Please see the line 37.

Comment 4: Line 45: Ensure consistency in the formatting of in-text citations (e.g., "Klemetsen et al. 2003" vs. "Kim et al. 2005").

Respond 4: The references in the manuscript were reformatted according to the journal guideline throughout this revised manuscript.

Comment 5: Line 50: Some sentences are long and complex, making comprehension difficult. Breaking them into shorter, more direct sentences would improve readability.

Respond 5: According to the comment we revised the paragraph to be more concise. Please see the lines 56-63.

Comment 6: Example: "As an aggressive predator and competitor, this alien species outcompetes native fish for food and habitat. It also preys on native aquatic animals, disrupting natural aquatic ecosystems."

Respond 6: According to the comment we corrected the sentence.

Comment 7: Line 62: The section states that "exact records of its introduction are unavailable." Providing more details on potential introduction pathways (e.g., aquaculture escape, illegal stocking) would strengthen the discussion.

Respond 7: There is no record for the introduction of S. trutta in South Korea. Instead, we slightly corrected the sentence as "Although its introduction has not been recorded in South Korea, ..."

Comment 8: Line 69: The temperature discussion appears somewhat contradictory. It states that most South Korean rivers approach 30°C in summer, which should hinder S. trutta establishment. However, the role of hydroelectric dam discharge in sustaining the species should be expanded upon to clarify how S. trutta persists in such conditions.

Respond 8: The sentence was corrected by providing the more detailed explanation for the water temperature maintenance by hydroelectric power generation. Please see the lines 77-78.

Comment 9: Line 113: While the authors modified existing primers, further validation steps, such as in silico specificity tests and empirical sensitivity assessments, should be included.

Comment 11: Line 221: The study provides detailed modifications of primers and probes, but a deeper discussion on the rationale behind these changes and how they improve efficiency over previous designs would enhance clarity.

Comment 12: Did the authors conduct an in silico analysis to assess specificity before empirical testing?

Respond 9, 11, and 12: Based on the aligned sequence matrix we modified the previous markers of Carim et al. (2016) to improve the PCR efficiency while guaranteeing their specificity. The characteristics of the modified markers were described in the Discussion section in detail. Please see the lines 312-326. However, this study, we did not conduct the comparison between the previous markers of Carim et al. (2016) and the present markers in this study, but we agree with the reviewer’s comment because such experiment will certainly improve the quality of this study.

Comment 10: Line 143: When describing the PCR mix, use "concentrations" instead of generic descriptions. Additionally, specify the amount of Taq polymerase (U) and buffer in the reaction mix.

Respond 10: We used GoTaq® Probe qPCR Master Mix (Promega, Madison, WI, USA) for this study. However, the Technical Manual does not provide information on the amount of Taq polymerase (U) and buffer in the reaction mix.

Comment 13: Line 248: The results indicate fluctuations in S. trutta eDNA across different sites and months. Expanding on potential environmental factors influencing these variations (e.g., water temperature, flow rate, degradation rates) would strengthen ecological insights.

Comment 16: Analyzing potential factors such as water temperature, flow rate, and eDNA degradation on detection patterns would strengthen ecological interpretations.

Respond 13 and 16: Unfortunately, we did not obtain such water parameters in this study. However, we agree with the reviewer’s comment because such environmental parameters will certainly nourish the discussion for this study and improve the quality of this study.

Comment 14: Some sites showed no S. trutta eDNA despite the species' known presence in the region. Addressing potential reasons for false negatives, such as eDNA degradation, dilution, or inhibition, would provide a more nuanced understanding of assay limitations.

Comment 17: Some locations lacked S. trutta eDNA despite known presence. Discussing potential causes such as sample degradation, dilution, or PCR inhibition would improve understanding of assay limitations.

Respond 14 & 17: However, we agree with the reviewer’s comment. In the future study, we have to consider internal and external amplification controls to trace the DNA degradation, dilution factor, and/or PCR inhibitors.

Comment 15: Line 280: Further explanation of how the detection limit relates to species abundance and ecological monitoring would be beneficial.

Comment 18: Incorporating this qPCR-based approach into continuous monitoring programs would enhance its practical application.

Respond 15 & 18: In our study, the detection limit of qPCR assay was 100 copies per reaction based on the standard curve. However, we have to say that the copy number does not explain the actual number of S. trutta individuals. Our qPCR data provide the probable sites of S. trutta occurrences and the insights for the further field survey in the future.

Comment 19: Overall, this study presents a valuable contribution to invasive species monitoring using eDNA. Addressing these points would further strengthen the manuscript and its applicability to conservation and management efforts.

Respond 19: We appreciate the reviewer’s invaluable comment.

Reviewer 4 Report

Comments and Suggestions for Authors

This study establishes a qPCR-assay to detect invasive brown trout based on eDNA samples in South Korea. This is very timely topic and I have only a few minor (non-technical) comments:

  1. introduction: Instead of starting the introduction with some general facts about salmonids, I strongly recommend starting (and in part replacing the first paragraph) wiht a general intro on the South Korean freshwater fish fauna an potential threats, including invasive species , which then would provide a logical link to your study.
  2. line 108, M&M: delete "salmonid" as you have also sampled some non-salmonid speceis
  3. M&M: I might have missed it, but did you confirm species-specificity of your qPCR assay by e.g. sequencing the amplified fragment in a few (also eDNA) samples?
  4. discussion, last paragraph: yes, S. trutta might and does hybridize with other salmonid species. But is there any evidence for hybridization with the species native to South Korea (maybe evidence from otehr countries/regions)?
Comments on the Quality of English Language

Though largely ok, the ms might benefit form one round of thorough proofreading (especially reagarding word use - sometimes better fitting words could be used; e.g., "occur" instead of "inhabit" as last word in the introduction, as one example.

Author Response

We sincerely thank the reviewer for the thoughtful and constructive comments that helped improve the quality and clarity of our manuscript. Your valuable feedback guided us in refining our work, and we appreciate the time and effort you dedicated to reviewing our paper.

Comment 1. Introduction: Instead of starting the introduction with some general facts about salmonids, I strongly recommend starting (and in part replacing the first paragraph) with a general introduction on the South Korean freshwater fish fauna potential threats, including invasive species, which then would provide a logical link to your study.

Respond 1: According to the other reviewer's comment the first paragraph was deleted to emphasize and focus on solely Salmo strutta.

Comment 2. Line 108, M&M: delete "salmonid" as you have also sampled some non-salmonid species

Respond 2: According to the suggestion we corrected the sentence. Please see the line 121.

Comment 3. M&M: I might have missed it, but did you confirm species-specificity of your qPCR assay by e.g. sequencing the amplified fragment in a few (also eDNA) samples?

Respond 3: In the "Result" section we described the confirmation of the positive amplifications of the qPCR assay by Sanger sequencing. Please see the lines 286-288.

Comment 4. Discussion, last paragraph: yes, S. trutta might and does hybridize with other salmonid species. But is there any evidence for hybridization with the species native to South Korea (maybe evidence from other countries/regions)?

Respond 4: There is no evidence for the hybridization between S. trutta and other salmonid species in South Korea. However, please see the lines 60-62 for the references of its hybridization reports in other countries.

Comment 5. Though largely OK, the ms might benefit form one round of thorough proofreading (especially regarding word use - sometimes better fitting words could be used; e.g., "occur" instead of "inhabit" as last word in the introduction, as one example.

Respond 5: According to the suggestion we changed the word. Please see the line 110.
